# Phenotypical and Functional Alteration of γδ T Lymphocytes in COVID-19 Patients: Reversal by Statins

**DOI:** 10.3390/cells11213449

**Published:** 2022-10-31

**Authors:** Marta Di Simone, Anna Maria Corsale, Elena Lo Presti, Nicola Scichilone, Carmela Picone, Lydia Giannitrapani, Francesco Dieli, Serena Meraviglia

**Affiliations:** 1Central Laboratory of Advanced Diagnosis and Biomedical Research (CLADIBIOR), AUOP Paolo Giaccone, 90127 Palermo, Italy; 2Department of Biomedicine, Neuroscience and Advanced Diagnosis (BIND), University of Palermo, 90127 Palermo, Italy; 3National Research Council (CNR), Institute for Biomedical Research and Innovation (IRIB), 90146 Palermo, Italy; 4Division of Respiratory Medicine, AUOP Paolo Giaccone, 90127 Palermo, Italy; 5Internal Medicine Department Unit, Health Promotion Sciences, Maternal and Infant Care, Internal Medicine and Medical Specialities Department (PROMISE), University of Palermo, 90127 Palermo, Italy

**Keywords:** γδ T cells, COVID-19, SARS-CoV-2 infection, statin, mevalonate pathway

## Abstract

(1) Background: statins have been considered an attractive class of drugs in the pharmacological setting of COVID-19 due to their pleiotropic properties and their use correlates with decreased mortality in hospitalized COVID-19 patients. Furthermore, it is well known that statins, which block the mevalonate pathway, affect γδ T lymphocyte activation. As γδ T cells participate in the inflammatory process of COVID-19, we have investigated the therapeutical potential of statins as a tool to inhibit γδ T cell pro-inflammatory activities; (2) Methods: we harvested peripheral blood mononuclear cells (PBMCs) from COVID-19 patients with mild clinical manifestations, COVID-19 recovered patients, and healthy controls. We performed ex vivo flow cytometry analysis to study γδ T cell frequency, phenotype, and exhaustion status. PBMCs were treated with Atorvastatin followed by non-specific and specific stimulation, to evaluate the expression of pro-inflammatory cytokines; (3) Results: COVID-19 patients had a lower frequency of circulating Vδ2+ T lymphocytes but showed a pronounced pro-inflammatory profile, which was inhibited by in vitro treatment with statins; (4) Conclusions: the in vitro capacity of statins to inhibit Vδ2+ T lymphocytes in COVID-19 patients highlights a new potential biological function of these drugs and supports their therapeutical use in these patients.

## 1. Introduction

The broad spectrum of clinical manifestations of COVID-19 disease depends on the interaction between SARS-CoV-2 and the immune system. In the majority of SARS-CoV-2-infected patients, innate and adaptive immune responses regulate the outcome of infection with limited and controlled inflammatory reactions. However, in some cases, the immune response becomes dysfunctional, causing severe pulmonary disease and overwhelming systemic illness, associated with an inflammatory syndrome with elevated serum levels of pro-inflammatory cytokines, profound lymphopenia, and coagulopathy in multiple vascular territories [1]. The panorama of medical therapies to treat COVID-19 is growing and evolving rapidly. Statins, traditionally administered to lower serum cholesterol, have been considered an attractive class of drugs in the setting of COVID-19 [2,3] due to their pleiotropic effects, including anti-inflammatory, immunomodulatory, antithrombotic, and antimicrobial properties [4,5]. Previously studies have evaluated the use of statins in the treatment of pneumonia and ARDS (Acute Respiratory Distress Syndrome) [6,7,8,9,10,11,12]. Particularly, an analysis of 540 individuals from the HARP-2 (hydroxymethylglutaryl-CoA Reductase Inhibition with Simvastatin in Acute Lung Injury to Reduce Pulmonary Dysfunction-2) trial demonstrated improved survival upon statin treatment in patients with a hyperinflammatory phenotype [11]. 

Statins suppress Toll-like receptors and cytokine (TNF-α, IL-1β, and IL-6) expression both in vitro and in vivo [13,14,15,16,17], and inhibit the NLRP-3 (nucleotide-binding domain, leucine-rich-containing family, pyrin-domain containing-3) inflammasome [18].

Furthermore, statins are well-known modulators of the adaptive immune system through their effect on lymphocytes and antigen-presenting cells [19,20]: they promote Th2 responses leading to the increased production of anti-inflammatory cytokines (IL-4, IL-5, and IL-10) [21] and inhibit IL-12-dependent Th1 cell differentiation and the production of pro-inflammatory cytokines IL-2, IL-12, IFN-γ, and TNF-α [22], which are believed to play an important role in the pathogenesis of COVID-19. Statins also promote the induction of endothelial nitric oxide synthase (eNOS), which improves endothelial functions and consequently reduces cell-adhesion-molecule expression and leukocyte chemotaxis [23,24,25,26,27,28,29].

The combined effects of statins on the innate and the adaptive immune systems could lead to the quenching of inflammation and decrease tissue damage and organ dysfunction. 

Here, we propose a further immunobiological mechanism of the therapeutic efficacy of statins, related to their inhibitory effect on γδ T cells functions, which contributes to inflammatory status and immunopathology. 

γδ T lymphocytes are important effector cells of the immune system that may play a role in bacterial and viral infections as well as in anti-tumor immunosurveillance. γδ T cells expressing the Vδ2 chain (preferentially paired to the Vγ9 chain) circulate in the peripheral blood and secondary lymphoid organs [30] and recognize small unprocessed non-peptidic compounds containing phosphate and termed phosphoantigens (PAgs), that are produced through the isoprenoid biosynthesis pathway [31]. Physiologic levels of PAgs, however, are not stimulatory of Vγ9Vδ2 T cells, but transformed and infected cells would produce increased metabolic intermediates such as PAgs [32,33,34]. Moreover, this pathway is easily manipulated by drugs, i.e., statins that are competitive and reversible inhibitors of 3-hydroxy-3-methylglutarylcoenzyme A (HMG-CoA) reductase, the key enzyme, which converts HMG-CoA into mevalonic acid, decreases PAg accumulation, and inhibits γδ T cell activation and their effector functions [35,36].

A growing number of studies have highlighted the contribution of unconventional T cells to both the pathogenesis and resolution of COVID-19 infection and disease, albeit with mechanisms not yet defined. Thus, this study aims to analyze γδ T lymphocytes in hospitalized and recovered COVID-19 patients to understand whether these cells contribute to the inflammatory status of COVID-19. Moreover, we have studied the in vitro effect of statins on the functional activity of Vδ2+ T cells, to highlight a new potential biological function of statins that supports their therapeutical use in COVID-19 patients.

## 2. Materials and Methods

### 2.1. Characteristics of Patient’s Cohort

We enrolled forty-eight patients affected by COVID-19, hospitalized between September and April 2021 at the Pneumology and Medicine Units of Paolo Giaccone University Hospital in Palermo, Italy. All patients showed mild clinical manifestations, and clinical laboratory data were registered at the time of the blood collection (Table 1). We did not also enroll patients with severe clinical manifestations in our study, because of the strong lymphopenia, which made it impossible to perform our analysis. In addition, 20 subjects who had recovered from COVID-19 for at least six months enrolled for hyperimmune plasma donation at the Immunohematology and Transfusion Medicine Department, and a cohort of 26 healthy subjects, were recruited and used as controls.

### 2.2. Analysis of Frequency, Phenotype, and Exhaustion Markers of γδ T Lymphocytes

Peripheral blood mononuclear cells (PBMCs) were separated from whole blood by density-gradient centrifugation using Ficoll-Paque PLUS (Cytiva). To assess the ex vivo frequency, phenotype, and exhaustion-marker expression of γδ T lymphocytes, PBMCs were stained with specific fluorochrome-conjugated monoclonal antibodies: APC-Cy7-conjugated anti-human CD45 (clone REA747), PerCP-Vio700-conjugated anti-human CD3 (clone REA613), PE-conjugate anti-human Vδ1 (clone REA173), PE-Vio770-conjugate anti-human Vδ2 (clone REA771), FITC-conjugate anti-human TIM3 (clone F38-2E2), APC-conjugate anti-human PD-1 (clone PD1.3.1.3), PE-Vio615-conjugate anti-human CD27 (clone REA499), and VioBlue-conjugated anti-human CD45RA (clone REA1047) (Miltenyi Biotec, Bergisch Gladbach, Germany). Following staining, the cells were incubated for 20 min at room temperature in the dark, and subsequently washed with FACS buffer (PBS, 4% FBS, 2 mM EDTA) and acquired.

### 2.3. Evaluation of Cytokine Production

PBMCs were cultured at a concentration of 5 × 10^5^ cells/mL in 24-well plates in RPMI 1640 supplemented with 25 mM HEPES buffer, 100 IU/mL penicillin, 0.1 mg/mL streptomycin, 2 mM L-glutamine, and 10% heat-inactivated FBS. To study the intracellular expression of IFN-γ, IL-17, and TNF-α by γδ T lymphocytes, PBMCs were stimulated for 6 h with ionomycin (1 μg/mL) and phorbol 12-myristate 13-acetate (PMA) (150 ng/mL) or for 48 h with zoledronate (2.2 μM/mL) and IL-2 (100 IU/mL), both in the presence of 10 μg/mL inhibitor of trans-Golgi function (Monensin) added 1 h after stimulation. Furthermore, to evaluate whether the inhibition of HMG-CoA reductase could affect cytokine production by γδ T lymphocytes, PBMCs were pre-treated for 24 h with atorvastatin at 10 µM/mL or 50 µM/mL concentration. After the incubation, the cells were harvested, washed with FACS buffer and surface stained using the following fluorochrome-conjugated monoclonal antibodies: APC-Cy7-conjugated anti-human CD45 (clone REA747), PerCP-Vio700-conjugated anti-human CD3 (clone REA613), FITC-conjugated anti-human Vδ1 (clone REA173), or Vδ2 (clone REA771) (Miltenyi Biotec). Intracellular cytokine expression was evaluated upon fixation and permeabilization using the Inside Stain Kit (Miltenyi Biotec) according to the manufacturer’s instructions. PE-conjugated anti-human TNF-α (clone REA656), PE-Vio615-conjugate anti-human IL-17-A (clone REA17A), and APC-conjugated anti-human IFN-γ (clone REA600) (Miltenyi Biotec) monoclonal antibodies were used to stain intracellular targets.

### 2.4. Flow Cytometric and Statistical Analysis

Samples were acquired at FACSAria (BD Biosciences, Franklin Lakes, NJ, USA) and analyzed by FlowJo software (Tree Star, version 10.5.3, Singapore, Singapore). The gating strategy is shown in Appendix A. Data were analyzed for statistical significance using the Mann–Whitney test for two groups, * *p* < 0.05, ** *p* < 0.01, *** *p* < 0.001, and **** *p* < 0.0001. Differences between groups with a probability of ≤0.05 were regarded as significant. All values are expressed as mean or median ± SD. 

## 3. Results

### 3.1. SARS-CoV-2 Infection Impaired Vδ1+ and Vδ2+ T Cell Frequency

As expected, COVID-19 patients with mild symptoms had marked lymphopenia and a reduction in the total T (CD3+) lymphocyte count. In addition, we found a statistically significant reduction in the ex vivo frequencies of circulating Vδ1+ and Vδ2+ T lymphocytes, as compared to healthy subjects. Interestingly, while the frequency of Vδ2+ T lymphocytes partially reversed 6 months after recovery, that of Vδ1+ T lymphocytes was still significantly decreased after recovery (Figure 1).

### 3.2. SARS-CoV-2 Infection Induced Changes in Vδ1+ and Vδ2+ T Cells Phenotype

We analyzed ex vivo the memory phenotype distribution of Vδ1+ and Vδ2+ T lymphocytes. As compared to healthy donors, Vδ1+ T lymphocytes from hospitalized COVID-19 patients had a statistically significant prevalence of the naive (CD45RA+CD27+) subset, accompanied by a significant reduction in the central memory (CD45RA-CD27+) subset and almost complete absence of the effector-memory (CD45RA-CD27−) and terminally-differentiated (CD45RA+CD27−) subsets (Figure 2A,B). Patients who had recovered from COVID-19 had a similar Vδ1+ subset distribution as those hospitalized patients. In contrast, Vδ2+ T lymphocytes from hospitalized COVID-19 patients had significantly predominant effector memory and terminally differentiated phenotypes, compared to healthy subjects, with a parallel reduction in naive and central memory phenotypes. This trend was maintained in COVID-19 patients 6 months after recovery (Figure 2C,D).

### 3.3. SARS-CoV-2 Infection Induces Expression of Exhaustion Markers on γδ T Cells 

To investigate the mechanisms underlying the decrease in Vδ1+ and Vδ2+ T lymphocytes in COVID-19 patients, we analyzed their exhaustion-marker expression. Compared to healthy subjects, Vδ1+ and Vδ2+ T lymphocytes from hospitalized COVID-19 patients had a statistically significant increase in the expression of the exhaustion markers PD-1 and TIM-3, either alone or in combination, measured both as a percentage of positive cells and mean fluorescence intensity (MFI), highly suggestive of their exhaustion status. This exhaustion profile of Vδ1+ and Vδ2+ T lymphocytes was maintained after the healing phase and was mainly sustained by TIM-3 expression. Percentages of Vδ1+ and Vδ2+ T lymphocytes expressing only PD1 or PD1 and TIM3 slightly decreased upon healing but were still significantly higher than the levels detected in healthy subjects. These data were fully paralleled by calculating the frequency of exhausted γδ T lymphocytes per million CD3+ cells (Figure 3).

### 3.4. SARS-CoV-2 Infection Induces Differential Cytokine Production by γδ T Cells

We assessed the relative frequencies of TNF-α-, IFN-γ-, and IL-17-producing Vδ1+ and Vδ2+ T cells at the time of infection. The frequency of these distinct populations defines the quality of the γδ T cell response. As compared to healthy donors, Vδ1+ T cells from COVID-19 patients spontaneously expressed TNF-α and IL-17, while unstimulated Vδ1+ T cells from recovered COVID-19 patients preferentially expressed IFN-γ, but also significant levels of TNF-α and IL-17. This cytokine expression pattern was even maintained upon Iono/PMA stimulation in vitro, which, it is of note, caused the only limited increase in cytokine expression (Figure 4). Despite their lower frequency in the peripheral blood of COVID-19 patients, Vδ2+ T lymphocytes spontaneously expressed IFN-γ both in hospitalized and recovered COVID-19 patients, as compared to healthy subjects. Upon Iono/PMA stimulation in vitro, Vδ2+ T lymphocytes from the three tested groups expressed IFN-γ, IL-17, and TNF-α, with the latter as the predominant cytokine (Figure 5). These data indicate that, although the disease reduced their frequency, Vδ1+ and Vδ2+ T lymphocytes have an activated pro-inflammatory profile suggesting their participation in COVID-19 pathogenesis.

### 3.5. Differential Cytokine Production of Vδ2+ T Cells upon Zoledronate Stimulation

Subsequently, in a cohort of 20 hospitalized COVID-19 patients, we evaluated cytokine expression by Vδ2+ T lymphocytes upon specifical stimulation with zoledronate. As shown in Figure 6, stimulation with zoledronate increased TNF-α, IFN-γ, and IL-17 expression in Vδ2 T lymphocytes from both healthy donors and COVID-19 patients, but Vδ2+ T lymphocytes from these latter expressed significantly more IFN-γ and IL-17 compared to healthy donors. In comparing cytokine production by Vδ2+ T lymphocytes from different groups, we noted that the MFI values were not identical, suggesting differences in the potency of the cytokine responses even among discrete populations of cells. These differences in the frequency and MFIs of Vδ2+ T cells prompted us to utilize a metric that incorporates both the magnitude and quality of response. The quality of a response is defined by the total frequency of Vδ2+ T cells producing a particular cytokine, whereas the MFI is used to assess the potency or magnitude of the response. By multiplying the frequency by the MFI, a metric is obtained and termed integrated MFI (iMFI), which reflects the total functional response of a population of cytokine-producing cells. iMFI values confirmed the exaggerated cytokine response in COVID-19 patients, as compared to healthy subjects (Figure 6).

### 3.6. Atorvastatin Inhibited Cytokine Production of Vδ2+ T Cells in COVID-19 Hospitalized Patients

Several studies have described the efficacy of statins as a therapeutic strategy to improve the outcome and prognosis of SARS-CoV-2 infection. Furthermore, by inhibiting the cholesterol biosynthesis pathway, statins also block p-Ag accumulation, resulting in γδ T lymphocyte inhibition. Therefore, we evaluated the in vitro effect of statins, in particular atorvastatin, on the cytokine response of Vδ2+ T lymphocytes It is noteworthy that zoledronate-induced Vδ2+ T cell cytokine production was inhibited by atorvastatin in a dose-dependent manner. In particular, we noticed a statistically significant inhibition of IFN-γ and IL-17 expression at both tested concentrations (10 and 50 μM/mL), while a significant reduction in TNF-α expression was achieved at the maximum tested dose (50 μM/mL) (Figure 7). The specificity of the inhibitory effect exerted by atorvastatin at the level of the mevalonate pathway was confirmed by the failure of atorvastatin to modify cytokine expression by non-specific stimulation with Iono/PMA (Figure 8).

## 4. Discussion

The progressive global pandemic caused by SARS-CoV-2 infection suggests that it is urgent to better understand the pathophysiology of COVID-19, given that approximately 20% of COVID-19 patients develop severe disease and 5% of them require intensive care. Severe disease is associated with organ damage, which not only involves the upper respiratory tract and lungs, but other organs as well [37,38]. Moreover, it is often related to alterations in immune activity, including increased levels of pro-inflammatory cytokines (e.g., IL-6 and TNF-α), lymphopenia, and T cell depletion [39,40,41]. 

During the 2003 epidemic, in patients with mild SARS-CoV infection, an orchestrated innate and adaptive immune response was shown to control and eradicate the infection. It was described that cell-mediated antiviral mechanisms against SARS-CoV involved both αβ and γδ T cells. Poccia et al. [42] described a selective expansion of Vδ2+ T cells, but not αβ T cells, with effector memory phenotype in peripheral blood of healthcare workers who recovered from SARS-CoV infection 3 months after disease onset, highly suggestive of their participation in the development of long-term immunity against SARS-CoV. Currently, few studies have included γδ T lymphocytes in the immune characterization of COVID-19 [43]. 

In the present study we evaluated the dynamics of γδ T lymphocytes in the peripheral blood of hospitalized and recovered COVID-19 patients compared to a cohort of healthy donors due to the lack of additional hospitalized patients with non-COVID-19 respiratory disease. In addition to a general condition of lymphopenia, we observed that the percentages of circulating total T and γδ T lymphocytes, as well as their Vδ1 and Vδ2 subsets, are significantly lower in COVID-19 hospitalized patients, compared to healthy controls. This was in line with the results obtained in other studies [44,45,46,47,48], particularly Laing et al. [45], who found that lymphopenia most overtly affected CD8 and γδ T cells, resulting in a significant reduction in circulating Vδ2 T cells accompanied by a substantial increase in Vδ1+ cells. Therefore, the depletion of the Vδ2 T cell subset in COVID-19 patients was severity-related and not shared with non-COVID-19 low respiratory tract infections and/or other diseases.

In contrast, other studies [49] did not find differences in the frequencies of γδ T lymphocytes between patients with COVID-19, non-COVID-19 patients, and healthy subjects; despite in the study by Carissimo et al. [50], both Vδ1+ and Vδ2+ T cells decreased with increasing disease severity [51].

Furthermore, another study combining flow cytometry and scRNA-seq analysis [52] showed a reduction in the number of γδ T lymphocytes in COVID-19 patients compared to healthy controls. Finally, the ratio between the absolute number of immature neutrophils and Vδ2+ (or CD8+) T lymphocytes predicted the onset of pneumonia and hypoxia with high sensitivity and specificity [44]. 

In our study, analysis of the memory phenotype of the γδ T cell compartment showed that Vδ2+ T lymphocytes had a dominant EM or TEMRA phenotype, but Vδ1+ T lymphocytes had a preferential naive phenotype, and these phenotypes were maintained after recovery. Unlike the COVID-19 patients recruited in our study, subjects who recovered from SARS-CoV-2 infection and were available for hyperimmune plasma donation showed partial recovery of the percentage of Vδ2+, but not of Vδ1+ T lymphocytes. Other studies have shown conflicting results about the differentiation stage of γδ T lymphocytes. Odak et al. described a marked increase both in frequency and an absolute number of naive γδ T lymphocytes and a decrease in those with effector phenotype in patients with moderate and severe COVID-19 [47]. Conversely, Carissimo et al. described a prevalence of TEMRA phenotype for Vδ1+ T lymphocytes in COVID-19 patients, which was maintained after recovery, and the prevalence of naive cells among Vδ2+ T lymphocytes, which instead increases in the recovered [44]. 

Given the condition of lymphopenia, which also affects circulating γδ T lymphocytes in these patients, it has been hypothesized that SARS-CoV-2 infection may induce a switch to a depleted phenotype or a condition of hyperactivation; this is supported by results that report overexpression of activation markers (CD38, HLA-DR, CD25) and genes encoding pro-apoptotic molecules (NFRSF10B/TRAIL-R and CASP3/Caspase 3) [45,53]. Indeed, in our cohort of COVID-19 patients, SARS-CoV-2 infection induced an increase in the expression of PD-1 and TIM-3 in both subsets of γδ T lymphocytes, as previously reported [49]. In contrast, in other studies, PD-1 expression did not differ between COVID-19 patients and healthy subjects [48] or between recovered and ICU patients [49]. A role of TIM-3 in the depletion of γδ T lymphocytes has also been described in patients with malaria, in whom high TIM-3 expression on Vδ2+ T cells is associated with reduced production of pro-inflammatory cytokines [54].

It is likely that the expression of exhaustion markers reflects their activation, rather than exhaustion, as also demonstrated for CD8+ SARS-CoV-2-specific T cells [55,56]. Indeed, upregulation of CD25 [48] and CD38 [44] expression is also observed in all differentiation stages of CD8+, Vδ1+, and Vδ2+ T lymphocytes in COVID-19 patients, except in Vδ2+ TEMRA, compared to recovered and healthy subjects [48,49]. On the other hand, no substantial alteration in γδ T lymphocyte CD69 expression is observed in patients with severe COVID-19 compared to non-COVID-19 patients, suggesting that this state of activation could be a general reflection of a severe condition, rather than a specific feature of SARS-CoV-2 infection [48,49].

Irrespective of the memory phenotype, both Vδ1+ and Vδ2+ T lymphocytes of COVID-19 patients spontaneously produced TNF-α, IFN-γ, and IL-17, which are sustained after healing. In addition, after non-specific Iono/PMA or specific zoledronate stimulation, Vδ2+ T lymphocytes expressed higher levels of TNF-α and IFN-γ in the presence of infection. Jouan et al. [49] previously studied cytokine production by γδ T lymphocytes during COVID-19 and found they produced less IFN-γ and more IL-17 after Iono/PMA stimulation, although levels detected were always low compared to healthy subjects. However, selective cytokine production by Vδ1+ and Vδ2+ T lymphocytes was not assessed. Hence, our data clearly show that during COVID-19, γδ T lymphocytes have a pro-inflammatory activity, which, in theory, might contribute to the cytokine storm. 

In our study, we evaluated whether the effector/pro-inflammatory activity of Vδ2+ T lymphocytes in hospitalized COVID-19 patients was inhibited by atorvastatin. In fact, statins, drugs with a cholesterol-lowering action, block the upstream enzyme hydroxy-methyl-glutaryl CoA reductase (HMG-CoA) of the mevalonate pathway and inhibit the production of PAgs, metabolites which specifically activate Vδ2+ T lymphocytes. Several studies suggest that statins may play a role in the treatment of SARS-CoV-2 infection, mainly by reducing complications due to pre-existing atherosclerotic cardiovascular disease and preventing acute lung injury [57,58,59]. Indeed, several observational clinical studies and meta-analyses have demonstrated a significant protective effect of statins in COVID-19 patients, administrated before or during hospitalization, reducing the risk of the need for invasive mechanical ventilation, mortality, ARDS, and hospital stay [3,60,61,62,63,64,65,66,67,68,69]. Overall, these studies suggest that this class of drugs may represent an important choice for adjuvant therapy in the treatment of these patients, influencing their course favorably. Considering this, several clinical trials are currently underway to evaluate the beneficial effects of statins in COVID-19 patients either when given alone or in combination with other drugs.

However, other studies suggest that statin use does not improve the prognosis of SARS-CoV-2-infected patients [70,71]. In addition, it could be possible that statin-treated patients may be at increased risk of SARS-CoV-2 infection and disease exacerbation, as this class of drugs has been shown to increase ACE2 receptor expression in mouse models [72,73]. 

It has been shown that the positive effect of statins on the mortality rate and prognosis of COVID-19 patients depends on direct and indirect antiviral mechanisms [64,67,74]. The first includes (a) the reduction in endogenous synthesis of cholesterol and, consequently, its content in the lipid rafts, limiting the penetration of the virus into the host cells [75], and (b) the inhibition of the endocytosis of the virus in the host cell [76] and (c) the inhibition of viral replication by acting on principal protease (Mpro) [77] and RNA-dependent RNA polymerase (RdRp) [78]. Indirect antiviral mechanisms, on the other hand, include (a) the reduction in the inflammatory process and cytokine storm, reducing IL-6 levels and inhibiting the Toll-like receptor (TLR) 4 and the TLR-MYD88-NF-кB pathway [64,79], (b) the attenuation of macrophage activation syndrome [64,79], (c) the improvement of endothelial function, reducing the production of reactive oxygen species and the pro-inflammatory activity of the NLRP3 inflammasome [64,79], (d) possible antithrombotic and antifibrotic activity [64,79,80], and (e) the increase in the concentration of arachidonic acid, interfering in ACE2/SARS-CoV-2 binding [81].

In our cohort of hospitalized COVID-19 patients, the activation of Vδ2+ T lymphocytes is inhibited by atorvastatin in a dose-dependent manner, resulting in a significant reduction in the expression of the pro-inflammatory cytokines TNF-α, IFN-γ, and IL-17. The same reduction was not observed after non-specific stimulation with Iono/PMA, suggesting that the inhibitory effect exerted by atorvastatin specifically impacts the mevalonate pathway. 

## 5. Conclusions

Altogether, our results demonstrate that circulating Vδ2+ T lymphocytes in COVID-19 patients, despite being present at a lower frequency, show an activated pro-inflammatory profile, suggesting they might contribute to the pathogenesis of COVID-19. Therefore, the in vitro ability of statins to inhibit the functional activity of Vδ2+ T lymphocytes in COVID-19 patients might provide an additional basis for the use of statins as an adjunct therapy in the treatment of patients with COVID-19.

## Figures and Tables

**Figure 1 cells-11-03449-f001:**
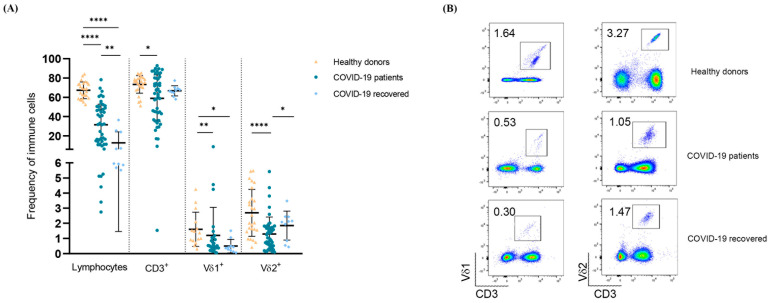
Frequency of Vδ1+ and Vδ2+ T lymphocytes in hospitalized and recovered COVID-19 patients and in healthy subjects. (**A**) Distribution of percentages of total lymphocytes, T (CD3+), Vδ1+ and Vδ2+ T lymphocytes in hospitalized and recovered COVID-19 patients, compared to healthy subjects. Shown is mean ± SD. (**B**) Representative dot-plots showing the frequency of Vδ1+ and Vδ2+ T lymphocytes in hospitalized and recovered COVID-19 patients and healthy subjects. * *p* < 0.05, ** *p* < 0.01, **** *p* < 0.0001.

**Figure 2 cells-11-03449-f002:**
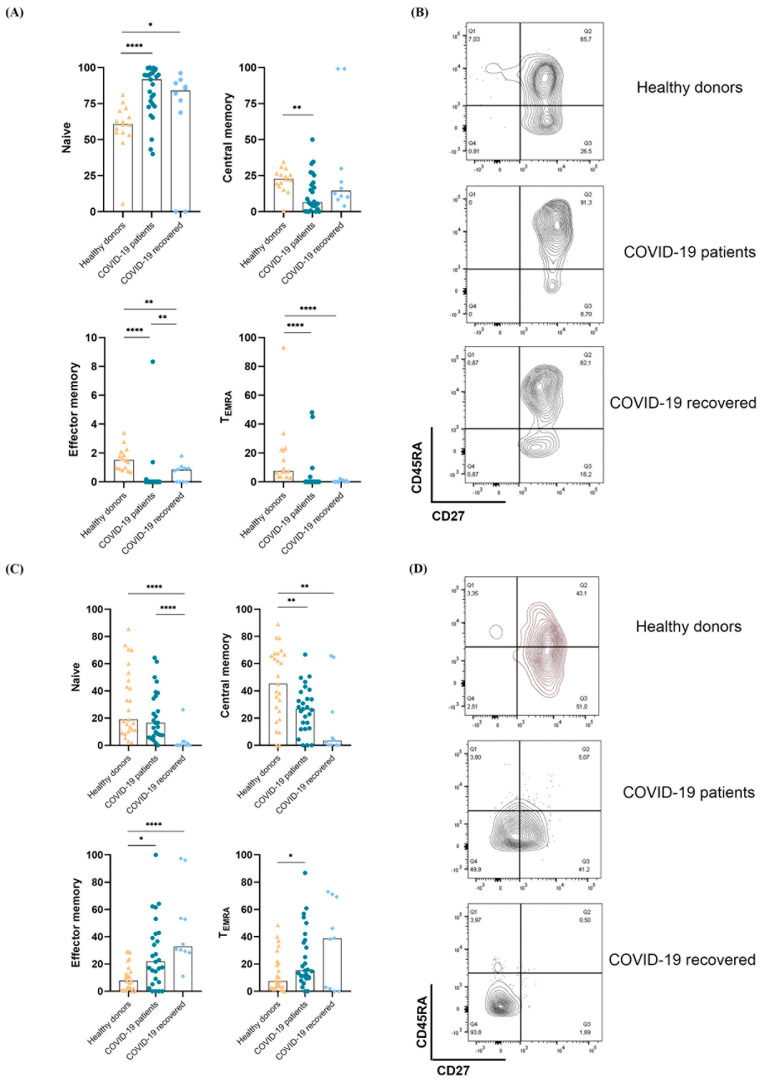
Phenotypic analysis of circulating Vδ1+ and Vδ2+ T lymphocytes in hospitalized and recovered COVID-19 patients and in healthy subjects. Distribution of ex vivo memory subsets of Vδ1+ (**A**) and Vδ2+ (**C**) T lymphocytes based on the expression of CD27 and CD45RA in hospitalized and recovered COVID-19 patients, compared to healthy donors. Median is shown. Healthy donors were represented as triangle, COVID-19 patients as circle and COVID-19 recovered as rhombus. Representative dot-plots showing Vδ1+ (**B**) and Vδ2+ (**D**) T lymphocyte memory subset distribution in hospitalized, recovered, and healthy subjects. * *p* < 0.05, ** *p* < 0.01, **** *p* < 0.0001.

**Figure 3 cells-11-03449-f003:**
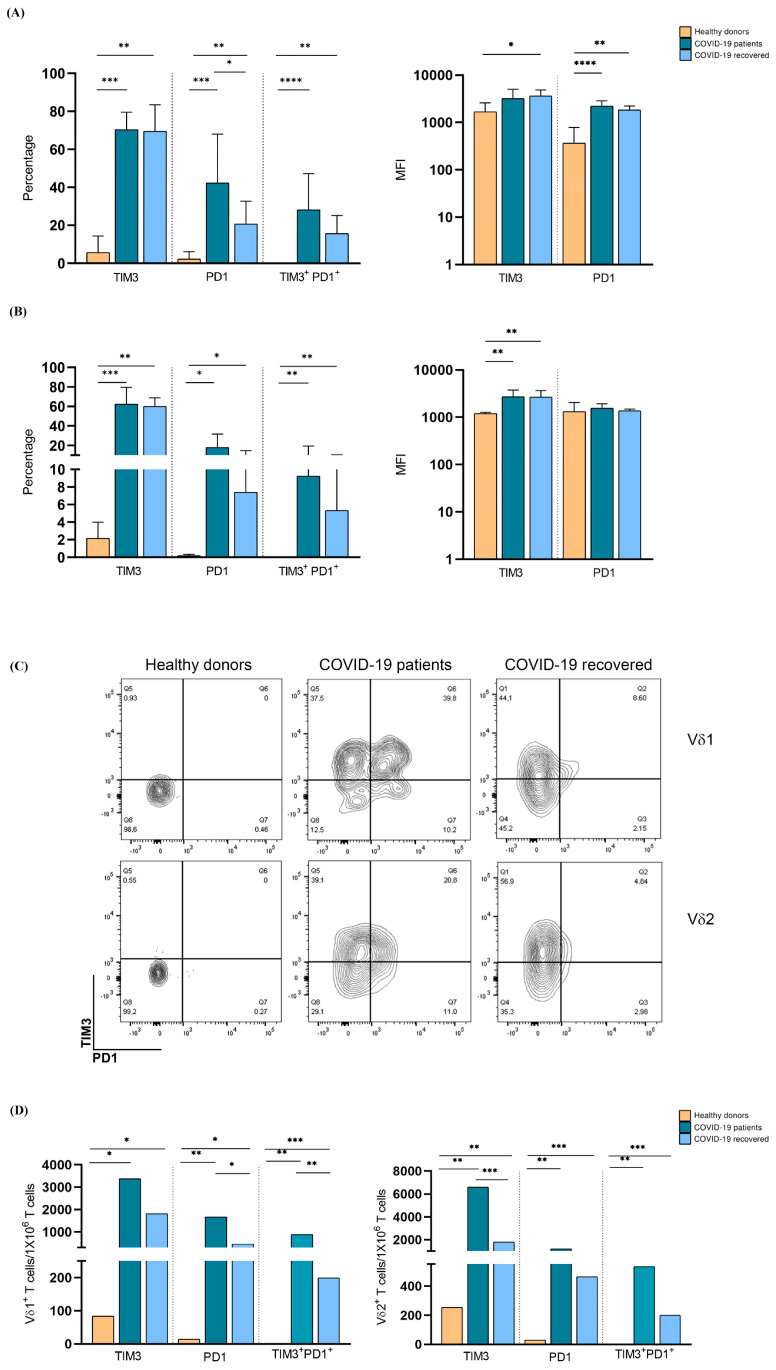
Analysis of exhaustion-marker expression by circulating Vδ1+ and Vδ2+ T lymphocytes in hospitalized and recovered COVID-19 patients and healthy subjects. Expression of TIM-3 and PD-1 on Vδ1+ (**A**) and Vδ2+ (**B**) T lymphocytes in hospitalized and recovered COVID-19 patients, compared to healthy donors. Shown are percentage (**left panels**) and MFI (**right panels**) ± SD. (**C**) Representative dot-plots of exhaustion-marker expression by Vδ1+ and Vδ2+ T lymphocytes from hospitalized and recovered COVID-19 patients, compared to healthy donors. (**D**) Frequency of exhausted Vδ1+ and Vδ2+ T cells per 1 × 10^6^ T lymphocytes. * *p* < 0.05, ** *p* < 0.01, *** *p* < 0.001, and **** *p* < 0.0001.

**Figure 4 cells-11-03449-f004:**
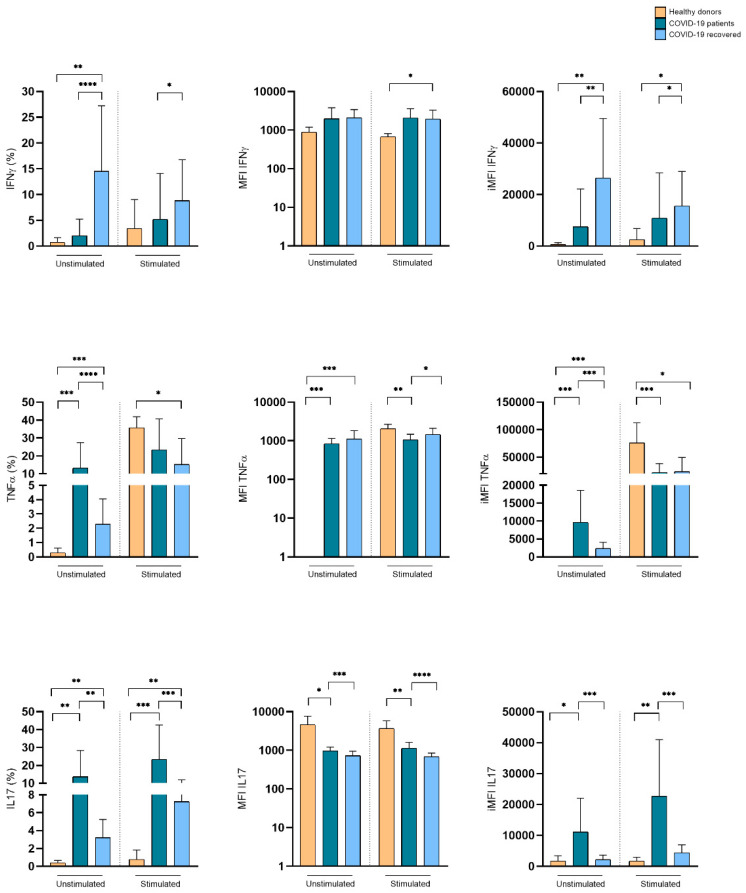
Expression of pro-inflammatory cytokines by Vδ1+ T lymphocytes in hospitalized and recovered COVID-19 patients, compared to healthy subjects. Cumulative histograms representing the expression of pro-inflammatory cytokines by Vδ1+ T lymphocyte. Shown are percentage (**left panels**), MFI (**central panel**, log_10_ scale), and iMFI (**right panels**) ± SD. * *p* < 0.05, ** *p* < 0.01, *** *p* < 0.001, and **** *p* < 0.0001.

**Figure 5 cells-11-03449-f005:**
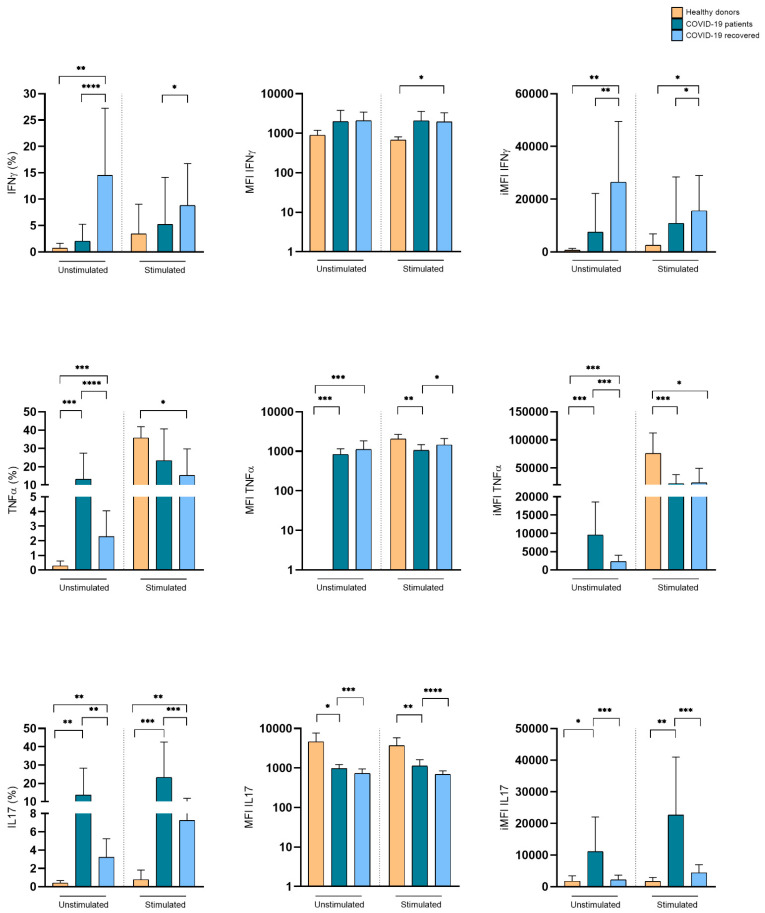
Expression of pro-inflammatory cytokines by Vδ2+ T lymphocytes in hospitalized and recovered COVID-19 patients, compared to healthy subjects. Cumulative histograms representing the expression of pro-inflammatory cytokines. Shown are percentage (**left panels**), MFI (**central panel**, log_10_ scale), and iMFI (**right panels**) ± SD. * *p* < 0.05, ** *p* < 0.01, *** *p* < 0.001, and **** *p* < 0.0001.

**Figure 6 cells-11-03449-f006:**
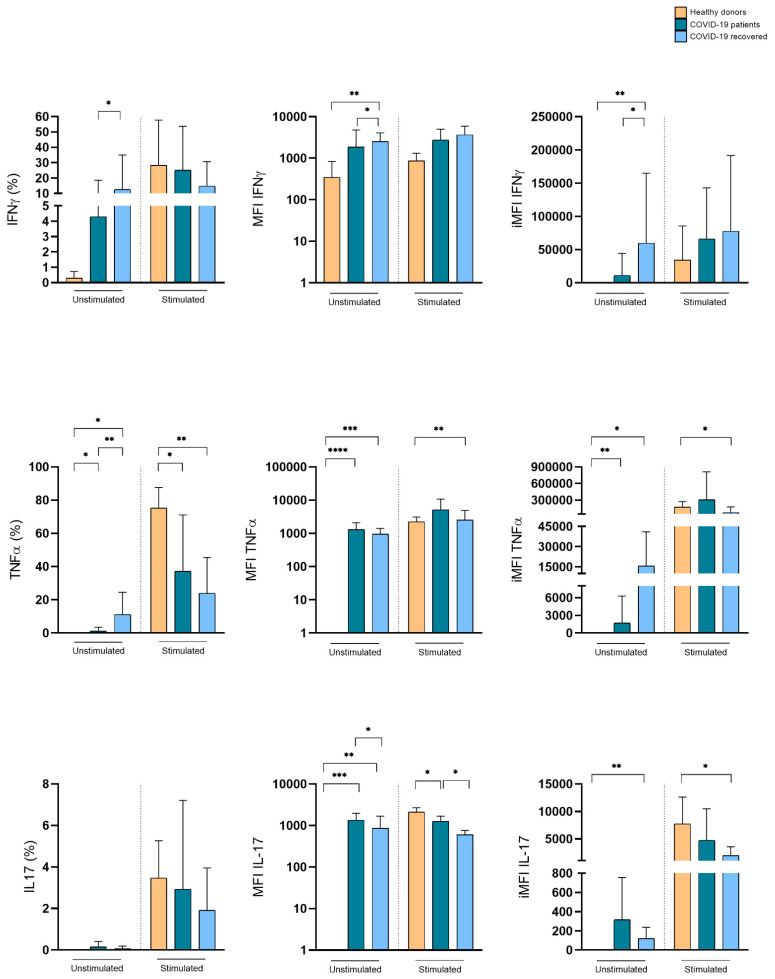
Intracellular cytokine expression by Vδ2+ T lymphocytes in COVID-19 patients and healthy donors upon zoledronate stimulation. Intracellular levels of IFN-γ, TNF-α, and IL-17, in terms of percentage and iMFI, expressed by Vδ2+ T cells, are represented as histogram plots. Shown is mean ± SD. * *p* < 0.05, ** *p* < 0.01, *** *p*< 0.001, and **** *p* < 0.0001.

**Figure 7 cells-11-03449-f007:**
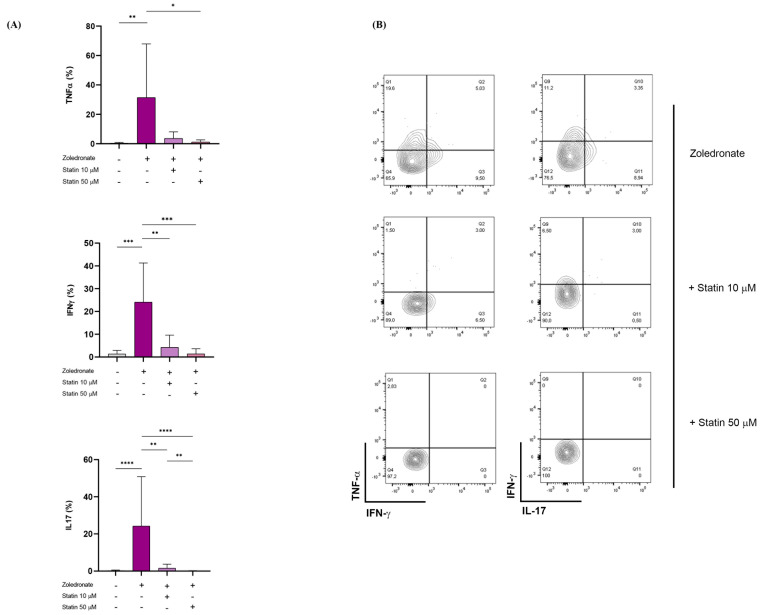
Inhibition of Vδ2+ T cell cytokine expression by atorvastatin upon zoledronate stimulation (**A**) Intracellular levels of TNF-α, IFN-γ, and IL-17 expressed by Vδ2+ T cells from COVID-19 patients upon stimulation with zoledronate in presence of atorvastatin. Data are represented as histogram plots. Each histogram shows mean ± SD. Purple histogram represented Zoledronate condition, lilac histogram represented Zoledronate + 10 μM statin condition, and pink histogram represented Zoledronate + 50 μM statin condition. (**B**) Representative counterplots showing pro-inflammatory cytokine expression by Vδ2+ T lymphocytes. * *p* < 0.05, ** *p* < 0.01, *** *p*< 0.001, and **** *p* < 0.0001.

**Figure 8 cells-11-03449-f008:**
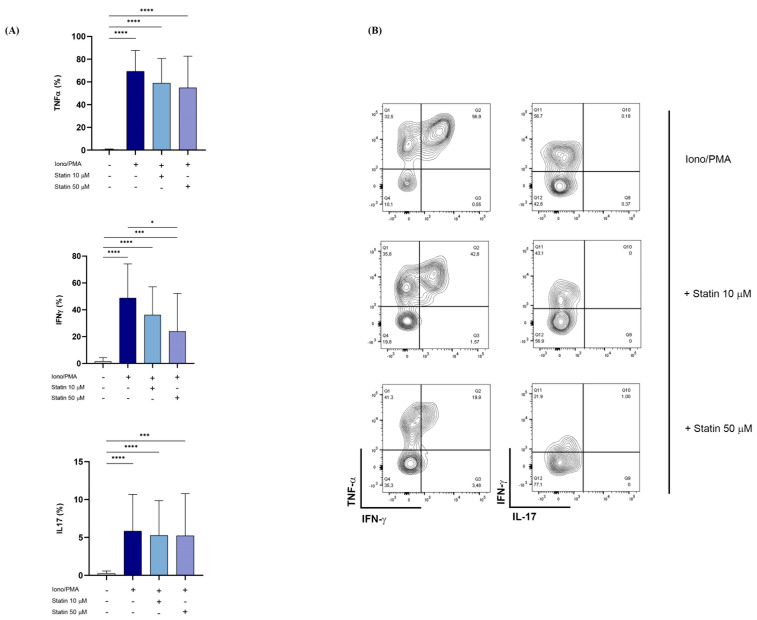
Inhibition of Vδ2+ T cell cytokine expression by atorvastatin upon Iono/PMA stimulation. (**A**) Intracellular levels of TNF-α, IFN-γ, and IL-17 expressed by Vδ2+ T cells from COVID-19 patients upon stimulation with Iono/PMA in presence of atorvastatin. Data are represented as histogram plots, showing mean ± SD. Blue histogram represented Iono/PMA condition, light blue histogram represented Iono/PMA + 10 μM statin condition, and light blue powder histogram represented Iono/PMA + 50 mM statin condition. (**B**) Representative counterplots showing pro-inflammatory cytokine expression by Vδ2+ T lymphocytes. * *p* < 0.05, *** *p*< 0.001, and **** *p* < 0.0001.

**Table 1 cells-11-03449-t001:** Demographic and clinical characteristics of the cohorts under study.

Demographic Characteristics	Hospitalized COVID-19 Patients	Recovered COVID-19 Patients	Healthy Donors
Number	48	20	26
Age means, range	63.91 (16–93)	42.2 (24–57)	28.9 (26–54)
Gender	♂34	♂15	♂11
	♀14	♀5	♀15
Laboratory data			
Leukocytes	8.84958	7490	6.811
(cells/mL)	(1.990–14.490)	(5.800–10.800)	(4.820–9.470)
Lymphocytes	1.6327	2.070	2.177
(cells/mL)	(130–13.200)	(1.500–3.200)	(1.030–3.039)
Neutrophils	8.12385	4.610	3.603
(cells/mL)	(1.170–74.200)	(2.900–7.400)	(2.060–5.720)
CRP (mg/mL)	74.9	/	1.40
n.v. <5 mg/mL	(0.16–3.380)		(0.23–4.95)
IL-6 (pg/mL)	18.72	/	1.91
n.v. <7 pg/mL	(1.5–118)		(1.55–6.83)

n.v: normal value.

## Data Availability

The data presented in this study are available on request from the corresponding author.

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
