# Peer review of "Phenotypical and Functional Alteration of γδ T Lymphocytes in COVID-19 Patients: Reversal by Statins"

_cells, 2022, doi:10.3390/cells11213449_

Round 1

Reviewer 1 Report

In this paper, Di Simone et al. investigate the effect of COVID on γδ T lymphocytes. They observe a decrease of these cells in patients with mild symptoms, with partial restoration in recovering participants. This change appeared to be associated with phenotypic alterations, decreased memory, and TEMRA cells increased exhaustion markers and effector functions. The author links these changes with COVID pathogenesis. Some of these changes were reversed in vitro by Atorvastatin, warranting the inclusion of statin in the treatment of COVID.

I appreciate that the authors looked into these rare (and rarely investigated) T cell subsets. I was intrigued by their findings, which could appeal to immunologists interested in the immunology of COVID disease. However, I have several concerns, questions, and comments, listed below.

Concerns:

-        The changes the authors report are connected to COVID pathogenesis. This should be nuanced, as their work does not provide any evidence that changes in γδ T lymphocytes have an actual impact on disease progression, severity, or recovery. Also, the design of their study includes COVID patients with mild disease. To connect with pathogenesis, the study should have included people with severe COVID.

-        Finally, the authors compare with “healthy donors.” How specific to COVID is their observation? How about other hospitalized patients with respiratory disease that were COVID-negative? If this cohort is not available, it should be noted as a limitation

-        Gating strategies for the gating of γδ T lymphocytes are lacking. Please provide.

-        Missing pieces of analysis missing are the frequency of exhausted or functional γδ T lymphocytes per million T cells. This is important because PD1, Tim3, TNFa, IFNy, and IL17 seem higher, but globally γδ T lymphocytes are down in COVID. It would be essential to know if these two paradoxical trends nullify each other.

-        In figure 2, how specific to γδ T lymphocytes are their findings? For example, do they see a similar trend for CD4 and CD8 T cells?

-        The authors acknowledge that PD-1 and Tim3 upregulation probably reflect a state of activation instead of exhaustion. In this context, I do not see the link between exhaustion marker expression and the decrease of γδ T lymphocytes. I did not see it discussed in the manuscript. It would have been helpful to test whether PD-1 or Tim3+ cells exert effector functions more or less.

-        In figure 3C, the gating of Tim3 is much more stringent than PD-1. What is the reason for that? With the gating strategy used, the authors seem to miss a low-grade upregulation an MFI analysis would have revealed (the summit of the contour plot is shifted by half a log in COVID+ participants). What was the problem of Tim3 expression in the Vδ2 contour plot in recovered COVID-19?

-        Stimulating with PMAionocymin and such a potent activating agent is not informative as it is far from physiologic. Interpretation of this data should be nuanced accordingly. PMAionomycin has a severe impact on specific markers such as CD4 and CD8. Can the authors confirm they still look especially at γδ T lymphocytes after such treatments?

-        The only cytokine contour plots come late in figure 7. They are problematic. On the left, IFNy is shown against TNFa. There are about 1.5% IFNy+ cells (Q2+Q3) with Zoledronate. In the right contour plot, assuming it is the same condition, there is about 15% IFNy+. The gate in the two plots is placed in the same place. Can the authors explain? It questions how these gatings were executed. A similar problem arises in figure 8.

-        The authors should assess the polyfunctional state of the γδ T lymphocytes.

-        The use of iMFI is misleading. It does not replace the iMFI. Instead, it helps appreciate the combined effect of frequency + MFI. Here, the MFI is missing. Therefore, it is impossible to interpret whether the global impact of function (defined by iMFI here) is caused by a change of frequency and/or cytokine expression per cell (shift in functionality).

-        The effect of statin is only seen after strong in vitro stimulation. I am not convinced that this experiment provides insight into what may happen in vivo. This is particularly problematic because the authors make a strong point about it. Why not extend the analysis to Vδ1 T lymphocytes?

-        The discussion is quite long, should be tightened, and speculation should be kept to a minimum. I do not understand the model they suggest. Zoledronate increases activation γδ T lymphocytes. Since the authors link activated γδ T lymphocytes to COVID pathogenesis, I would think, following their reasoning, that Zoledronate would be a bad thing in the context of COVID. I guess the most straightforward strategy would be to avoid Zoledronate rather than use a second drug (statin) to cancel the effect of Zoledronate. Unless Zoledronate is used for another reason, I fail to understand. It also causes a problem for the title, as it infers that statins reverse the alteration of the γδ T lymphocytes during COVID. Statin counters the effect of  Zoledronate, not COVID.

Minor comments.

-        Please add the axes in Figure 1 B.

-        Some are odd in figure 2A. The mean histogram for recovered participants is off. The use of mean may cause it, but it misrepresents the real distribution. I recommend median instead.

-        Please use log scale or log-transformed data for MFI.

Reviewer 2 Report

1. Correct some typos and punctuation errors, as well as some linguistic inaccuracies. 2. obtain a paragraph within the discussion section that can also take into consideration results obtained "in vitro" and not only "in vivo" on different organs in the course of Covid-19. From a search on Pubmed / Medline I suggest the following manuscripts as, in my opinion, the paper will be improved further.
von Massow G, Oh S, Lam A, Gustafsson K. Gamma Delta T Cells and Their Involvement in COVID-19 Virus Infections. Front Immunol. 2021 Oct 29;12:741218. doi: 10.3389/fimmu.2021.741218. PMID: 34777353; PMCID: PMC8586491.
Geng J, Chen L, Yuan Y, Wang K, Wang Y, Qin C, Wu G, Chen R, Zhang Z, Wei D, Du P, Zhang J, Lin P, Zhang K, Deng Y, Xu K, Liu J, Sun X, Guo T, Yang X, Wu J, Jiang J, Li L, Zhang K, Wang Z, Zhang J, Yan Q, Zhu H, Zheng Z, Miao J, Fu X, Yang F, Chen X, Tang H, Zhang Y, Shi Y, Zhu Y, Pei Z, Huo F, Liang X, Wang Y, Wang Q, Xie W, Li Y, Shi M, Bian H, Zhu P, Chen ZN. CD147 antibody specifically and effectively inhibits infection and cytokine storm of SARS-CoV-2 and its variants delta, alpha, beta, and gamma. Signal Transduct Target Ther. 2021 Sep 25;6(1):347. doi: 10.1038/s41392-021-00760-8. PMID: 34564690; PMCID: PMC8464593.
Resta L, Vimercati A, Cazzato G, Fanelli M, Scarcella SV, Ingravallo G, Colagrande A, Sablone S, Stolfa M, Arezzo F, Lettini T, Rossi R. SARS-CoV-2, Placental Histopathology, Gravity of Infection and Immunopathology: Is There an Association? Viruses. 2022 Jun 18;14(6):1330. doi: 10.3390/v14061330. PMID: 35746801; PMCID: PMC9227044.
Cazzato G, Colagrande A, Cimmino A, Cicco G, Scarcella VS, Tarantino P, Lospalluti L, Romita P, Foti C, Demarco A, Sablone S, Candance PMV, Cicco S, Lettini T, Ingravallo G, Resta L. HMGB1-TIM3-HO1: A New Pathway of Inflammation in Skin of SARS-CoV-2 Patients? A Retrospective Pilot Study. Biomolecules. 2021 Aug 16;11(8):1219. doi: 10.3390/biom11081219. PMID: 34439887; PMCID: PMC8392002.

Round 2

Reviewer 1 Report

I would like first to acknowledge the effort provided by the authors to improve the manuscript. A number of my concerns were elevated, some important remains.

Figure 3D should be logscale (or log transformed), with error bars.

I understand the focus is on γδ T cells. My suggestion to analyse a more global subset was to put their finding of the  γδ T cells in perspective, a perspective still lacking. I raise this issue because the authors make the point that γδ T cells seem to be involved in COVID19. Anything showing a specific effect of γδ T cells vs other lymphocytes would help make the point. Any disease can cause fluctuation in a subset of lymphocytes. It does not mean that these fluctuations are pathogenic.

Exhaustion does not necessarily lead to cell death, at least of other lymphocytes I know better. The authors did not prove exhaustion of γδ T cells (in fact, they themselves acknowledged upregulation of Tim3 and PD1 can be caused by the stimulation in vitro, which does not imply exhaustion in vivo), and did not prove that exhaustion can lead to γδ T cells decrease in number. Therefore, their hypothesis is overly speculative as formulated.

PMAionomycin was widely used in the past, but now is regarded as an improper tool to study lymphocytes because it forces functions that would not normally be exerted. I am not an expert on γδ T cells, but an effort from the authors to explain why  they cannot do otherwise will help. Showing a gating strategy ± PMAionomycin would certain help, as PMAionomycin is also known to mess with lineage markers.
